# Transcriptomic analysis reveals cloquintocet-mexyl-inducible genes in hexaploid wheat (*Triticum aestivum* L.)

**Olivia A. Landau**[ORCID][☯][¤][*], **Brendan V. Jamison**[☯], **Dean E. Riechers**[☯]

Department of Crop Sciences, University of Illinois, Urbana, Illinois, United States of America

☯ These authors contributed equally to the work.
¤ Current Address: U.S. Department of Agriculture-Agricultural Research Service, Wheat Health, Genetics, and Quality Research Unit, Pullman, Washington, United States of America
* olivia.landau@USDA.gov

## Abstract

Identification and characterization of genes encoding herbicide-detoxifying enzymes is lacking in allohexaploid wheat (*Triticum aestivum* L.). Gene expression is frequently induced by herbicide safeners and implies the encoded enzymes serve a role in herbicide metabolism and detoxification. Cloquintocet-mexyl (CM) is a safener commonly utilized with halauxifen-methyl (HM), a synthetic auxin herbicide whose phytotoxic form is halauxifen acid (HA). Our first objective was to identify candidate HA-detoxifying genes via RNA-Seq by comparing untreated and CM-treated leaf tissue. On average, 81% of RNA-Seq library reads mapped uniquely to the reference genome and 76.4% of reads were mapped to a gene. Among the 103 significant differentially expressed genes (DEGs), functional annotations indicate the majority of DEGs encode proteins associated with herbicide or xenobiotic metabolism. This finding was further corroborated by gene ontology (GO) enrichment analysis, where several genes were assigned GO terms indicating oxidoreductase activity (34 genes) and transferase activity (45 genes). One of the significant DEGs is a member of the CYP81A subfamily of cytochrome P450s (CYPs; denoted as *CYP81A-5A*), which are of interest due to their ability to catalyze synthetic auxin detoxification. To investigate *CYP* expression induced by HM and/or CM, our second objective was to measure gene-specific expression of *CYP81A-5A* and its homoeologs (*CYP81A-5B* and *CYP81A-5D*) in untreated leaf tissue and leaf tissue treated with CM and HM over time using RT-qPCR. Relative to the reference gene (β-tubulin), basal *CYP* expression is high, expression among these *CYPs* varies over time, and expression for all *CYPs* is CM-inducible but not HM-inducible. Further analysis of *CYP81A-5A*, such as gene knock-out, overexpression experiments, or *in vitro* activity assays with purified enzyme are necessary to test the hypotheses that the encoded CYP detoxifies HA and that CM upregulates this reaction.

**Data availability statement:** Illumina RNA-Seq data are available on NCBI (BioProject Accession #: PRJNA983309; https://www.ncbi.nlm.nih.gov/bioproject/983309). All other relevant data is included in the Supporting Information files.

**Funding:** This research was supported by an Undergraduate Research Award at the University of Illinois Urbana-Champaign to B.V.J. and funding from Corteva Agriscience to D.E.R. The funders had no role in study design, data collection and analysis, decision to publish, or preparation of the manuscript.

**Competing interests:** The authors have declared that no competing interests exist.

## Introduction

The allohexaploid *Triticum aestivum* L. (wheat) genome (2n = 6x = 42; AABBDD) contains 21 homologous pairs of chromosomes spread among three homoeologous sets of seven chromosomes [1,2]. Like other cereal crops, wheat is tolerant to the synthetic auxin herbicides, which are commonly utilized for selective postemergence dicot weed control [3,4]. The primary mechanism behind selectivity is qualitative and quantitative differences in detoxification of these herbicides between grasses and dicots [4]. The general process for plant herbicide metabolism can be divided into three parts: Phase I, Phase II, and Phase III. Phase I typically involves oxidation reactions catalyzed by cytochrome P450s (CYPs), and these metabolites are subjected to Phase II conjugation reactions with endogenous substrates (e.g., glucose, reduced glutathione, or amino acids) [5–8]. During Phase III, Phase II metabolites are transported into the vacuole by transport proteins (e.g., ATP-binding cassette (ABC) transport proteins), and within the vacuole metabolites are detoxified further or incorporated into the cell wall [6–9]. Synthetic auxin herbicides exemplify of how differential herbicide metabolism allows for herbicide selectivity between grass crops and dicot weeds. Generally, grasses possess CYPs that catalyze irreversible ring-hydroxylation or dealkylation reactions of synthetic auxin herbicides, forming a less toxic compound and predisposing the herbicide to glucose conjugation by UDP-dependent glucosyltransferase (UGTs) and subsequent sequestration to the vacuole by ABC transport proteins [6,9,10]. By contrast, synthetic auxin herbicide metabolism in dicots mainly consists of reversible reactions, such as amino acid or glucose conjugation of the carboxylic acid, which results in some level of the phytotoxic form of the herbicide remaining in the cell [3,11].

While the role of CYPs in herbicide detoxification in tolerant crops and resistant weeds has been established [6,12,13], a specific gene encoding a synthetic auxin detoxifying CYP has not been characterized in wheat. However, several examples of CYPs governing tolerance to certain herbicides have been documented in grass crop [14–23] and weed species [19,24–29], including synthetic auxins in the case of maize (*Zea mays*) *CYP81A9* [16]. These findings indicate members of the CYP81A subfamily are likely candidate genes associated with synthetic auxin detoxification.

The herbicide examined in this study, halauxifen-methyl (HM), was commercialized in 2015 and is a member of the 6-aryl-picolinic acid subclass of synthetic auxins [30,31]. HM is typically applied postemergence to wheat in tank mixtures with other herbicides [32]. Wheat is tolerant to HM through rapid detoxification of the phytotoxic form, halauxifen acid (HA) [33]. More specifically, once HM is de-esterified to HA, the HA is *O*-demethylated, and subsequently conjugated with glucose [33]. Previous HM phenotyping experiments [34] and results from liquid chromatography-mass spectrometry (LC-MS) experiments [35] indicate HA-detoxifying genes are located on chromosome 5A. Thus, further examination through RNA-Seq is warranted to identify candidate HA-detoxifying genes.

To prevent injury in wheat, cloquintocet-mexyl (CM) is typically included with postemergence applications of HM [32,33]. Safeners, like CM, are commonly applied with foliar herbicides to large-seeded cereals to reduce herbicide injury, which is accomplished by inducing expression and activity of herbicide detoxification and transporter enzymes [8,36,37]. While the phenotypic and metabolic effects of safeners are well documented, knowledge of safener regulation of corresponding genes or signaling pathways being induced for crop protection is still severely limited [7]. To date, transcriptome analyses of safeners in grass crops are relatively rare [38–42], and currently only one published paper has reported wheat transcriptomic data in response to safener, mefenpyr-diethyl [43]. Furthermore, gene expression induced by safeners implies the encoded enzymes may play a role in herbicide metabolism

[7–9,44]; however, further biochemical studies are required to functionally validate the function of candidate genes. Thus, the first objective of this research is to identify candidate genes responsible for HA-detoxification via RNA-Seq by comparing the expression of genes in untreated (UT) and CM-treated tissue. The second objective is to validate CM-inducible gene expression of candidate gene(s) and to compare expression of candidate gene(s) in response to CM and/or HM treatments via TaqMan RT-qPCR.

## Materials and methods

### Chemicals and plant materials

Chemicals used in the following experiments were provided by Corteva Agriscience and include CM (formulated as a 25% active ingredient wettable powder) and the Elevore™ formulation of HM (contains 6.87% HM and equates to 68.5 grams halauxifen acid equivalent per liter of Elevore™). Seed for the winter wheat variety, 'Kaskaskia' [45], was provided by Dr. Frederic Kolb at the University of Illinois at Urbana-Champaign.

### Seed sowing, treatment application and tissue collection

For both RNA-Seq and RT-qPCR experiments, seeds were planted in $382\,cm^3$ pots containing a 1:1:1 soil mixture of soil, peat, and sand. Pots were placed in a greenhouse room with a 14-hour day length and a constant 21 to 23°C temperature band. Natural light was supplemented with halide lamps delivering $800\,\mu mol\,m^{-2}\,s^{-1}$ photon flux to the plant canopy. Until the application of treatments, plants were watered at the soil surface once a day until the soil was uniformly moist. When seedlings produced 1–2 leaves (Zadoks stages 11–12), treatments were applied using a compressed air research sprayer calibrated to deliver $187\,L\,ha^{-1}$ at 275 kPa with an even flat-fan nozzle.

For the RNA-Seq experiment UT plants were sprayed with a 0.1% solution of nonionic surfactant (NIS), while CM-treated plants were sprayed with a solution containing 15 g a.i. $ha^{-1}$ of CM and 0.1% NIS [46]. After application of treatments, plants were returned to the greenhouse room. At 6 hours after treatment (HAT) the first leaves were cut at the collar from five plants (approximately 500 mg of leaf tissue) were collected, frozen with liquid nitrogen, and stored in a −80°C freezer until RNA extraction. This experiment was independently conducted three times with one UT sample and one CM sample from each experimental replication utilized for RNA-Seq sample submission. In total, three independent biological replicates for each treatment were submitted for RNA-Seq.

For the RT-qPCR experiment, treatments from the RNA-Seq experiment were included along with the addition of 5 g a.e. $ha^{-1}$ of HM and a combination treatment of CM and HM (CM+HM). All treatments included 0.1% NIS. Harvesting procedures were the same as previously mentioned but performed at 3, 6, and 12 HAT. This experiment was conducted three times. One biological replicate per treatment per timepoint from each experimental replication was utilized for TaqMan RT-qPCR. In total, three independent biological replicates for each treatment at a given timepoint were utilized for TaqMan RT-qPCR.

### RNA extraction, library construction, and transcriptomic analysis

Total RNA was isolated using previously described methods [34], and RNA concentration and purity were determined with a NanoDrop 1000 spectrophotometer (Thermo Scientific, USA). RNA samples with concentrations above $100\,ng/\mu L$, $A_{260}/A_{280}$ ratios above 1.8, and $A_{260}/A_{230}$ ratios between 2.0 and 2.3 were used in downstream processes. Each RNA sample (10 μg) was treated with TURBO™ DNase using the TURBO DNA-free™ Kit (Thermo

Scientific, USA) using the manufacturer's protocol to eliminate genomic DNA contamination. The concentration of DNase-treated RNA was determined with a Qubit 2.0 fluorometer (Invitrogen, USA) using the manufacturer's protocol.

Six RNA samples (three UT biological replicates and three CM-treated biological replicates) were submitted to the Roy J. Carver Biotechnology Center for the construction of RNA-Seq libraries. An AATI Fragment Analyzer was used to evaluate integrity of the RNA samples, which indicated the 28S and 16S bands were prominent and degradation was not detected. Libraries were quantitated by qPCR and sequenced on one lane for 151 cycles from each end of the fragments on a NovaSeq 6000, generating 150-bp paired end reads. RNA-Seq data quality was estimated by FastQC v0.12.0 (https://www.bioinformatics.babraham.ac.uk/projects/fastqc/) with low quality sequences filtered with fastp v0.17.0 [47]. None of the reads failed to pass the filter (Table 1). The 'Chinese Spring' reference genome (IWGSC RefSeq v1.1) and functional gene annotations were downloaded from URGI (https://wheat-urgi.versailles.inra.fr/). Salmon/1.10.1 was used to align clean reads to the reference genome and quantify reads [48]. All tests of significance for differential expression were analyzed with modules from the edgeR package [49] in R (version 4.2.0) using RStudio (Version 2023.03.0), in which the TMM normalization was utilized to adjust expression values to a common scale. Gene expression levels were calculated using counts-per-million and transformed to $\log_2$ counts per million. Significant DEGs were identified if fold changes in expression were ≥5 or ≤−5 and the false-discovery rate was <0.1. Illumina RNA-Seq data are available on NCBI (BioProject Accession #: PRJNA983309; https://www.ncbi.nlm.nih.gov/bioproject/983309). Molecular functions of DEGs were assigned gene ontology (GO) annotations with agriGO v2.0, which were used to perform GO enrichment analysis with default settings [50,51]. Of the 103 significant DEGs, 99 (96.1%) were assigned GO terms. The coding sequences of significant CYPs and UGTs were compared to Phytozome BLAST results (https://phytozome-next.jgi.doe.gov/) of CYP coding sequences maize [52] and rice (*Oryza sativa*) [53] using Phytozome BLAST (https://phytozome-next.jgi.doe.gov/).

## RNA preparation, primer design, thermal cycling conditions and data analysis for TaqMan RT-qPCR

RNA was prepared, evaluated for quality, and quantitated utilizing the same methods described in the previous section. However, instead of utilizing an AATI Fragment Analyzer

**Table 1. RNA-Seq libraries prepared from untreated (UT) and cloquintocet-mexyl (CM)-treated wheat leaf tissue.**

| Libraries | Total Number of Reads | QC Filtered | Unmapped | Mapped | Multi-mapped | Not in a Gene | Ambiguous | In a Gene |
|---|---|---|---|---|---|---|---|---|
| CM 1 | 136,557,553 | 0 | 5,368,294 | 110,013,166 | 12,859,553 | 13,336,028 | 79,316 | 104,914,362 |
| | | | (3.9%) | (80.6%) | (9.4%) | (9.8%) | (0.1%) | (76.8%) |
| CM 2 | 119,779,163 | 0 | 3,186,940 | 98,014,375 | 11,218,910 | 11,125,246 | 77,605 | 94,170,462 |
| | | | (2.7%) | (81.8%) | (9.4%) | (9.3%) | (0.1%) | (78.6%) |
| CM 3 | 105,647,929 | 0 | 3,170,235 | 85,508,314 | 11,064,971 | 9,901,715 | 70,150 | 81,440,858 |
| | | | (3.0%) | (80.9%) | (10.5%) | (9.4%) | (0.1%) | (77.1%) |
| UT 1 | 141,508,909 | 0 | 3,812,761 | 112,971,361 | 17,611,365 | 14,792,471 | 96,697 | 105,195,615 |
| | | | (2.7%) | (79.8%) | (12.4%) | (10.5%) | (0.1%) | (74.3%) |
| UT 2 | 111,036,055 | 0 | 3,505,913 | 88,539,503 | 12,354,153 | 11,249,162 | 78,751 | 83,848,076 |
| | | | (3.2%) | (79.7%) | (11.1%) | (10.1%) | (0.1%) | (75.5%) |
| UT 3 | 103,575,175 | 0 | 3,268,617 | 83,600,223 | 12,168,875 | 9,170,966 | 70,206 | 78,896,511 |
| | | | (3.2%) | (80.7%) | (11.7%) | (8.9%) | (0.1%) | (76.2%) |

to evaluate integrity, total RNA was examined for quality after denaturation at 55°C in the presence of formamide and formaldehyde, then integrity of rRNA bands were visualized with ethidium bromide in a 1.2% agarose gel containing 0.4 M formaldehyde [54].

Due to the high sequence identity (≥96%) among genes of interest, TaqMan RT-qPCR methodology was chosen to achieve homoeolog discrimination. TaqMan primers and probes were designed with AlleleID 7 (PREMIER Biosoft, USA) for the candidate gene TraesCS5A02G394800.1 and its homoeologs TraesCS5B02G402800.1 and TraesCS5D02G407300.1 (denoted as *CYP81A-5A*, *CYP81A-5B*, and *CYP81A-5D*, respectively; S1 and S2 Tables). Primers and probe were also designed for a β-tubulin gene (*β-TUB*; TraesCS7D02G454200.1) to serve as a reference gene (S1 and S2 Tables). This gene was chosen based on previous results indicating stable expression in wheat leaf tissue [55]. PCR efficiencies of *CYPs* and *β-TUB* primers were calculated in SDS 2.3 software (Applied Biosystems, USA) with six-point standard curves in a 10-fold dilution series of RNA (S1 Table).

RT-qPCR was conducted using a 7900 HT Sequence Detection System (Applied Biosystems, USA) and reactions were performed in 20 μL volumes, consisting of 10 μL TaqMan RT-PCR Mix (2x), 1.0 μL TaqMan Gene Expression Assay (20x), 0.5 μL TaqMan RT Enzyme Mix (40x), and 8.5 μL (170 ng) RNA template (TaqMan™ RNA-to-Ct™ 1-Step Kit; Applied Biosystems, USA). The following program was used for RT-qPCR: 48°C for 15 minutes, 95°C for 10 minutes, followed by 40 cycles at 95°C for 15 seconds and 60°C for 1 minute. Each sample was analyzed in three technical replicates and mean cycle threshold (Ct) values were calculated. Reverse-transcription negative controls were included to verify genomic DNA contamination was not contributing to Ct values. CM- and HM-induced gene expression for each *CYP* gene was calculated relative to transcript levels in the nontreated control samples (per treatment and timepoint) and normalized using *β-TUB* a reference gene with the $2^{-\Delta\Delta Ct}$ method.

Comparisons of the ΔΔCt and Ct values of genes were performed with the lme4 package (Bates et al., 2015) in R (version 4.2.0) using RStudio (Version 2023.03.0). A mixed effects model was utilized for both experiments where gene, treatment, and their interactions were treated as fixed effects and replicates were treated as random effects. Data was subjected to ANOVA and means were separated with Fisher's protected LSD (α = 0.05).

## Results

### RNA-Seq and GO analysis of differentially expressed genes

Among all RNA-Seq libraries for UT and CM-treated leaf tissue, 81% of reads mapped uniquely to the reference genome, 76.4% of reads were mapped to a gene, 10% of reads were not mapped to a gene, 3% of reads were unmapped, 11% of reads were multi-mapped, and 0.1% of reads were ambiguous (Table 1). In terms of significant differentially expressed genes (DEGs), 101 genes were induced and two genes were repressed by CM (Fig 1). Based on their functional annotations, the majority of these DEGs encode proteins associated with Phase I, II or III herbicide/xenobiotic metabolism (81 genes upregulated and 1 gene downregulated) while the remaining DEGs are associated with stress/defense response (2 genes upregulated and 1 gene downregulated), amino acid metabolism (5 genes upregulated), heavy metal binding (3 genes upregulated), or encode transcription factors (3 genes upregulated; Fig 2). The two repressed genes, TraesCS3A02G042700 and TraesCS7D02G370400, are annotated as a peroxidase (potentially associated with Phase I herbicide/xenobiotic metabolism) and a "negative regulator of resistance" (a stress/defense response gene), respectively (Figs 1 and 2; S3 Table). These results indicate gene repression by CM is minimal in leaf tissue at 6 HAT, and we hypothesize that CM triggers a certain level of oxidative stress—possibly to fine-tune

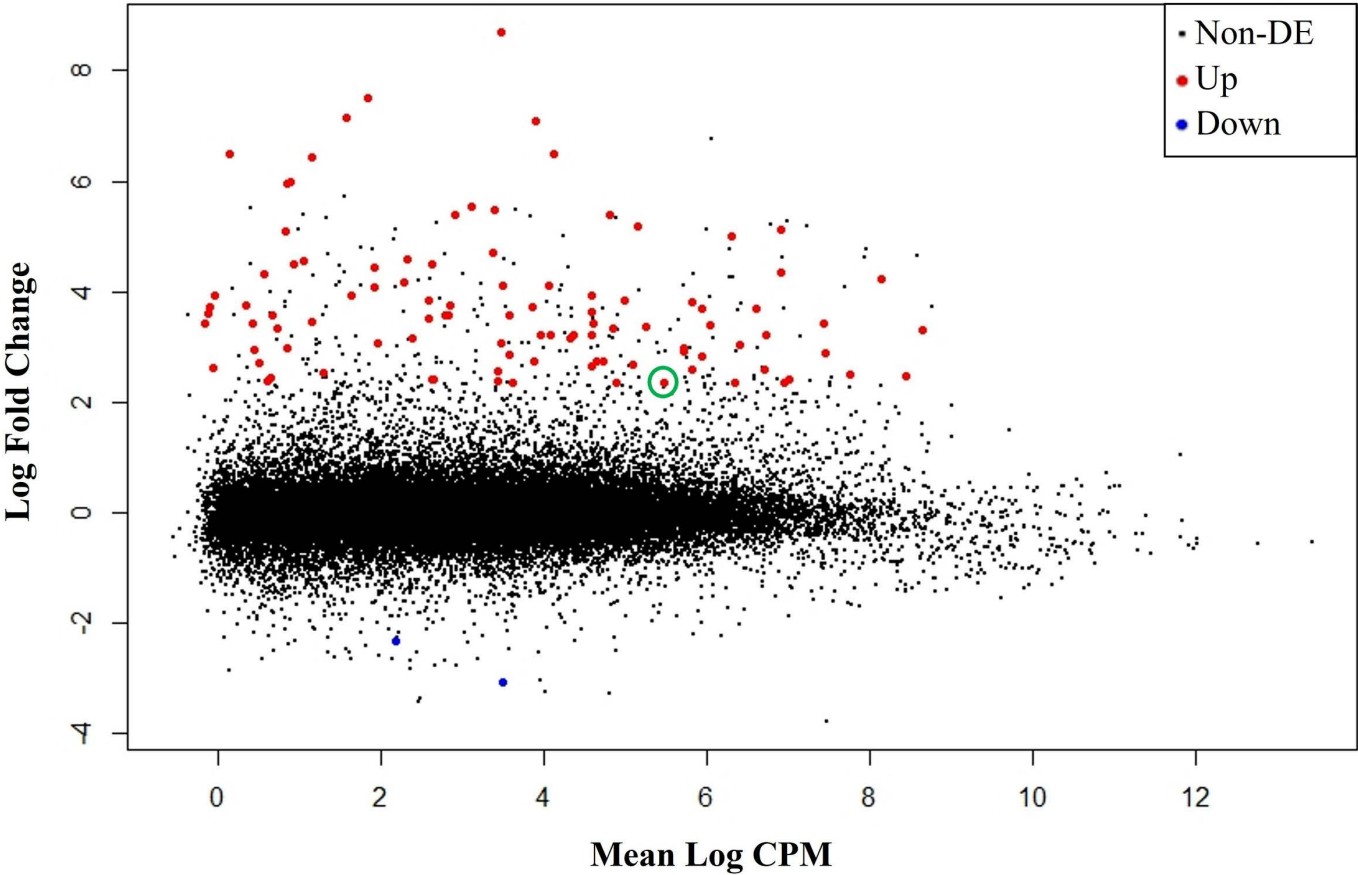

**Fig 1. Mean-difference plot showing the log$_2$ fold change and mean abundance of each transcript in log$_2$ counts per million (CPM).** Genes that were significantly induced (Up) and repressed (Down) by cloquintocet-mexyl are highlighted in red and blue, respectively. Genes that were not significantly differentially expressed (Non-DE) are highlighted in black. TraesCS5A02G397800.1 is circled in green.

safener-mediated transcriptional regulation. Overall, these results indicate CM treatment at 6 HAT primarily induces gene expression with a relatively minor degree of gene repression.

The GO enrichment analysis identified 34 genes significantly associated with oxidoreductase activity and 45 genes significantly associated with transferase activity, including 11 genes associated with glutathione *S*-transferase (GST) and 19 genes associated with UGT activity (S1 and S2 Figs). Results of GO enrichment analysis corroborate with the results of the tree map (Fig 2) and further elaborate on the function of the encoded proteins of the DEGs by indicating which substrates or molecular bonds are potentially involved in the reactions catalyzed by the proteins. For example, the GO terms assigned to a specific gene may not only indicate it is a UGT, but also could catalyze glucose conjugations with abscisic acid or indole-3-acetic acid (S4 Table).

Based on the results of previous LC-MS [35] and phenotypic experiments [34], we hypothesze major candidate gene(s) encoding HA-detoxifying enzyme(s) are located on wheat chromosome 5A. Of the 103 significant DEGs, five genes (three UGTs and two CYPs) are located on the group 5 chromosomes with fold inductions for these genes ranging from approximately 5 to 23 (S3 Fig). The UGTs are TraesCS5D02G404200.1, TraesCS5A02G394800.1, and TraesCS5B02G305600.1, while the CYPs are TraesCS5A02G472300.1 and TraesCS5A02G397800.1.

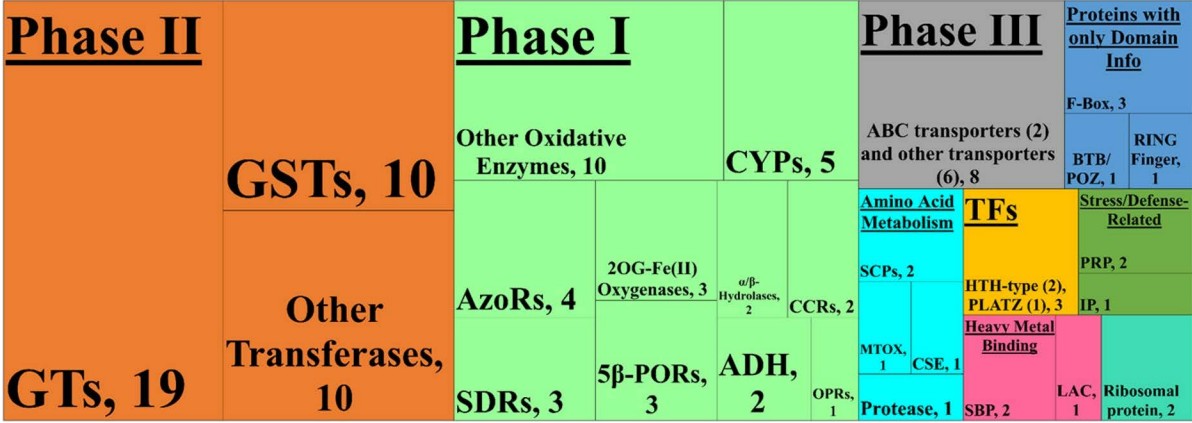

**Fig 2. Tree map of functional annotations assigned to significant differentially expressed genes identified by RNA-Seq.** The main categories (bolded and underlined) include Phase I (light green), Phase II (orange), and Phase III (grey) metabolism, Amino Acid Metabolism (light blue), Transcription Factors (TFs; yellow), Proteins with only Domain Info (dark blue), and Stress/Defense Related (dark green). The number of identified genes is listed in parentheses or after the comma. Abbreviations: 2OG, 2-oxoglutarate; 5β-PORs, progesterone 5-beta-reductase; ABC, ATP-binding cassette; ADH, alcohol dehydrogenase; AzoR, azoreductase; CCR, cinnamoyl-CoA reductase 4; CSE, cystathionine gamma-lyase; CYP, cytochrome P450; GST, glutathione *S*-transferase; GT, glycosyltransferase; HTH, helix-turn-helix; IP, inhibitor protein; LAC, laccase; MTOX, *N*-methyl-L-tryptophan oxidase; OPR, 12-oxophytodienoate reductase; PLATZ, plant AT-rich protein and zinc-binding protein; PRP, pathogen-related protein; SBP, selenium-binding protein; SCP, serine carboxypeptidase; SDR, short chain dehydrogenase/reductase.

Phytozome BLAST results of TraesCS5A02G472300.1 and TraesCS5A02G397800.1 display high sequence identity to members of CYP71C (≥70%) and CYP81A (≥80%), respectively. Both CYPs were assigned GO:0003824 (catalytic activity), GO:0016491 (oxidoreductase activity), and GO:0046914 (transition metal ion binding) (S4 Table), which are typical terms associated with CYPs. Based on resources from the International Wheat Genome Sequencing Consortium [2,56], the UGTs, TraesCS5D02G404200.1 and TraesCS5A02G394800.1, are homoeologs (94% identity), and Phytozome BLAST results indicate they have high sequence identity to members of UGT85A in maize and rice (≥71%). Both UGTs were assigned 23 GO terms specifying the glucosyltransferase activities for several compounds, including indole-3-acetate, benzoic acid, salicylic acid, abscisic acid and flavonol (S4 Table). Literature regarding the maize and rice UGT85As is not currently available, but other reports indicate that the UGT85As are in involved with plant stress responses in wheat, tobacco (*Nicotiana tabacum*), grapevine (*Vitis vinifera* × *Vitis labrusca*) [57–59]. TraesCS5B02G305600.1 shows high similarity to a salicylic acid glucosyltransferase 1 (SGT1) in rice (73% identity), which catalyzes the formation of glucoside and glucose esters of salicylic acid [60]. This UGT was assigned the same specific glucosyltransferase GO terms as the previous two UGTs, except for the terms associated with cytokinin, cis-zeatin, and hydroquinone activity (S4 Table). Furthermore, the assigned GO terms of GO:0052639 (salicylic acid glucosyltransferase (ester-forming) activity) and GO:0052640 (salicylic acid glucosyltransferase (glucoside-forming) activity) would suggest that TraesCS5B02G305600.1 is capable of catalyzing similar reactions associated with SGT1 (S4 Table).

With the exception of TraesCS5A02G472300, the other significant *CYP* and *UGTs* located on chromosome 5 (TraesCS5D02G404200, TraesCS5A02G394800, TraesCS5B02G305600, and TraesCS5A02G397800) are located in tandem gene clusters encoding similar enzymes (S5 Table). Gene clusters for TraesCS5A02G397800 and TraesCS5B02G305600 are relatively small in number with 3 other genes, while TraesCS5A02G394800 and its homoeolog

TraesCS5D02G404200 exist in clusters with 9 to 10 other genes (S5 Table). It is common for the wheat genome to contain tandemly duplicated genes [61]. Likewise, genes encoding metabolic enzymes, such as CYPs and UGTs, commonly exist in tandem gene clusters in plants and they can range in number (approximately 3–15) [62–67]. Current results indicate that genes within these clusters vary in terms of CM induction since only the one gene within these clusters was identified in this research (S5 Table).

Given the previous LC-MS [35] and phenotypic results [34] along with the wealth of research regarding CYP81A involvement in herbicide detoxification in plants [13,24,27,29], TraesCS5A02G397800.1 and its homoeologs, TraesCS5B02G402800.1 and TraesCS5D02G407300.1 (denoted as *CYP81A-5A*, *CYP81A-5B,* and *CYP81A-5D*, respectively) were selected for further analysis via gene-specific RT-qPCR.

### Expression of *CYP81A-5A, CYP81A-5B*, and *CYP81A-5D*

Overall, expression levels and fold inductions for the CYPs fluctuated throughout the time-course, and all three CYPs are CM-inducible but not HM-inducible (Fig 3). The CM and CM+HM treatments significantly induced expression of all CYPs at 3 HAT and 12

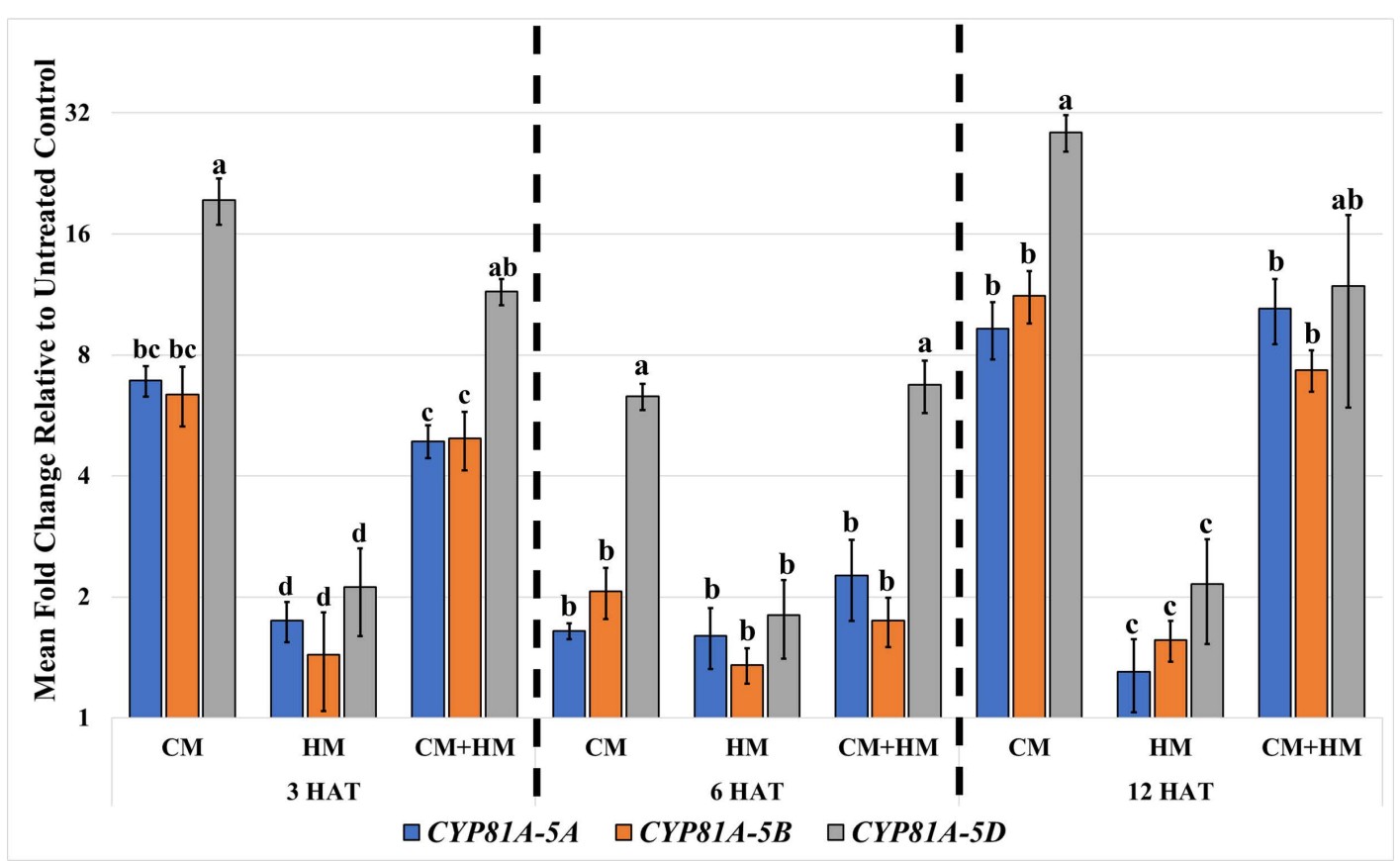

**Fig 3. Mean fold changes for *CYP81A-5A, CYP81A-5B*, and *CYP81A-5D* in response to cloquintocet-mexyl (CM) and halauxifen-methyl (HM) at 3, 6, and 12 hours after treatment (HAT).** Treatments include untreated control (0.1% nonionic surfactant (NIS)), CM (15 g a.i. ha⁻¹ of CM), HM (5 g a.e. ha⁻¹ of HM) and CM+HM (15 g a.i. ha⁻¹ of CM and 5 g a.e. ha⁻¹ of HM). All treatments included 0.1% NIS. Within each timepoint values that share the same letter are not significantly different ($\alpha$ = 0.05). Fold inductions for each gene at each timepoint were calculated by $2^{(-\Delta\Delta Ct)}$ with β-tubulin (*β-TUB*) as a reference gene. Mean fold changes represent results from three biological replicates (*n* = 3). Error bars represent standard error of the mean.

HAT where fold inductions ranged from approximately 4.8 to 28.6 (Fig 3). In contrast, HM treatments resulted in small (approximately 2.1-fold or less) fold changes in expression at all timepoints (Fig 3). At each timepoint, a CYP induced by CM was also induced by CM+HM and their fold inductions were not significantly different from each other (Fig 3). These results indicate a lack of additive or synergistic effects from the HM treatment. At 6 HAT, only the expression of *CYP81A-5D* is significantly induced by CM and CM+HM with both treatments resulting in an approximate 6.3- and 6.7-fold induction, respectively (Fig 3). Additionally, expression of all CYPs was always detected in UT tissue, especially at 3 HAT where all CYPs were not significantly different from *β-TUB* (S4 Fig). These results indicate *CYP81A-5A* expression is relatively high in UT tissue and, therefore, is likely required for yet-to-be identified endogenous function(s) at this growth stage in wheat.

## Discussion

### CM-regulated genes associated with herbicide detoxification

We identified 103 DEGs whose expression is significantly altered (101 genes were induced and two genes were repressed) by CM treatments. The DEGs are associated with varying stages of herbicide metabolism, stress/defense response, amino acid metabolism, heavy metal binding, or transcription factors (Figs 1 and 2; S3 Table), which is further corroborated by GO enrichment analysis (S1 and S2 Figs; S4 Table). As mentioned previously, published transcriptomes for safener-treated grass species is rare; however, similar trends exist pertaining to the types of significant DEGs identified between the current and previous results. Regardless of safener or species, genes encoding herbicide-detoxifying enzymes, such as CYPs, UGTs, GSTs, and ABC transport proteins were in identified in both current and previous research [38–43]. While these enzymes are commonly associated with herbicide detoxification, other genes encoding enzymes capable of catalyzing Phase I oxidative and Phase II conjugation reactions were also identified (S3 Table). Other Phase I oxidative enzymes included 2-oxoglutarate/Fe(II)-dependent oxygenase superfamily proteins, 12-oxophytodienoate reductases, short-chain dehydrogenase/reductases [38,42], and peroxidases [42], while other Phase II conjugation enzymes included HXXXD-type acyl-transferase family proteins [38]. While other reports of transcriptomic data for CM-treated wheat are unavailable, CM-treated tissues were examined via Northern blot and protein expression analysis in wheat [46,68] and *Aegilops tauschii* [54,69], the D genome progenitor of wheat. In each case, GSTs were significantly induced by CM [54,68,69], and two GSTs identified by Xu et al. [54] (denoted as *TtGSTU1* and *TtGSTU2*) were also identified in current results (Table S3). Differences in significant DEGs between current results and the results of previous experiments could be due to multiple differences in the experimental design, such as the choice of species, cultivar, safener, tissue type, plant growth stage, sampling timepoints, etc., could all influence results of transcriptome analysis. The effects of some of these factors have been explored in non-safener-related experiments in wheat [70,71], but the full extent of how these factors affect safener-induced gene expression has yet to be explored at the transcriptomic and proteomic levels.

CYPs and UGTs were anticipated as potential DEGs in the transcriptome data due to their common involvement in synthetic auxin herbicide detoxification [7,11,72,73] and their expression often induced by safeners [38–43]. CYPs are noteworthy due to their demonstrated ability to detoxify multiple classes of herbicides, especially in members of Poaceae: a plant family with the highest number of reported herbicide-detoxifying CYPs [13,27–29]. Given that members of the CYP81A subfamily commonly catalyze herbicide detoxification reactions [15–20,24–28,74], choosing to investigate *CYP81A-5A* in more detail was a reasonable hypothesis. However, members of CYP71C are also capable of

herbicide-detoxification [23] and increased expression has been reported for nicosulfuron-tolerant *Zea mays* [75] and cyhalofop-butyl-resistant *Leptochloa chinensis* [76]. Thus, TraesCS5A02G472300.1 could also be examined in the future for roles in herbicide detoxification. Overall, further exploration into herbicide-detoxification capabilities of wheat CYPs is warranted since there are limited examples, including biochemical evidence of CYP involvement in diclofop-methyl detoxification [77–79], and wheat CYP71C6V1 detoxifying of several acetolactate synthase-inhibiting herbicides via an *in vitro* yeast assay [23].

To add the complexity of potential xenobiotic substrates, natural substrates of herbicide-detoxifying CYP81As are currently unknown [13], and this information could enhance the understanding of what herbicide characteristics determine its potential of being a CYP81A substrate. Given that CYPs are typically involved in the biosynthesis of both primary and secondary metabolites, such as sterols, fatty acids, carotenoids, and hormones [80–82], there are many potential candidates for CYP81A natural substrates.

The significant CM-induced UGTs identified by RNA-Seq also warrant examination in the future, especially since reactions catalyzed by CYPs may result in phytotoxic metabolites and still require further detoxification by UGTs [6,13]. In general, herbicide-detoxifying UGTs are not as well characterized compared to CYPs, especially in the Poaceae, but their expression is often induced by herbicides and safeners [38,44,83,84]. Some examples in wheat include detection of glucosylated metabolites of isoproturon [85] and florasulam [86]. In rice four UGTs are implicated in the detoxification of 2,4-D, and inhibitors of photosystem II, 4-hydroxyphenylpyruvate dioxygenase, and very-long-chain fatty acid elongases [87–90]. Lastly, a UGT in *Alopecurus myosuroides* was implicated in conferring cross-resistance to several herbicides [91].

### *CYP81A-5A*, *CYP81A-5B*, and *CYP81A-5D* are CM-inducible but not HM-inducible

While the RT-qPCR experiment indicated that the expression of all examined CYPs is CM-inducible, expression between the three homoeologs varies over time and their expression is not equally induced by CM (Fig 3). Unequal homoeolog expression is not an unexpected result since expression patterns commonly vary among homoeologs in polyploids [70,92]. While *CYP81A-5D* displayed the largest inductions by CM across all timepoints (Fig 3), our previous results [34,35] indicate that *CYP81A-5D* alone is not sufficient for HA detoxification and that *CYP81A-5A* should be examined in future experiments. These differences between transcriptomic and phenotypic results could be partly explained by the generally weak correlation between mRNA transcript abundance and their corresponding proteins (usually around 40% in eukaryotes) [93]. We hypothesize that while *CYP81A-5D* has higher CM-inductions or comparable expression of *CYP81A-5A*, not all the *CYP81A-5D* transcripts are translated to functional, active proteins or the structural differences between the encoded proteins negatively affects HA binding for CYP81A-5D. Verifying increased protein abundance for any of the three gene products would require immunoblotting or ELISA assays.

The fluctuations in basal expression and CM inductions (Fig 3 and S4 Fig) during the time course are also not unexpected because many plant biological activities show diurnal variation, and the circadian clock coordinates plant activities in response to environmental cues, such as light and temperature [94]. Additionally, nearly all the genes associated with stress signaling pathways are regulated by the circadian clock, which synchronizes them for improved fitness and optimized development [95–97]. The objective of this experiment was not to measure the effect of the circadian clock on target genes; to address this objective,

reference genes lacking variation over a time course experiment would be required [98] or absolute quantification of transcripts could be performed by making standards to extrapolate CYP transcript abundance in samples. The *β-TUB* gene would not be a suitable reference gene for this objective because it displays variation between timepoints (S4 Fig).

### Future experiments and implications for *CYP81A-5A*

Given the results of previous experiments [34,35] and the abundant evidence of CYP81A involvement in the detoxification of multiple herbicides [16–19,29,99], *CYP81A-5A* will be examined for future herbicide detoxification characterization. If *CYP81A-5A* also catalyzes detoxification of multiple herbicides, it would have great potential for genetic transformation of relevant crop genomes lacking natural herbicide tolerance. Genes encoding herbicide detoxification enzymes could be utilized for *in vitro* metabolism assays with *E. coli* or yeast cells [24,100] to screen and predict whole-plant tolerance to numerous experimental herbicides in wheat prior to performing whole-plant phenotyping.

Functional validation of *CYP81A-5A* with clustered regularly interspaced short palindromic repeats (CRISPR)/CRISPR associated protein 9 (Cas9) methods is desired, which has been successfully utilized in wheat [101–104]. We therefore hypothesize CRISPR/Cas9-mediated modifications in *CYP81A-5A* that result in either a knockout or altered expression will have a commensurate effect on natural or CM-induced HM tolerance and possibly tolerance to other wheat-selective herbicides. When the function of *CYP81A-5A* has been validated, further exploration into its transcriptional regulation, expression in other tissues, and substrate binding will be performed.

## Conclusions

RNA-Seq identified 103 DEGs whose expression can be significantly altered (101 genes were induced and two genes were repressed) by CM treatments. These genes and encoded proteins are associated with varying stages of herbicide metabolism, stress/defense response, amino acid metabolism, heavy metal binding, or transcription factors. Further investigation with RT-qPCR indicated that *CYP81A-5A* expression is CM-inducible but not HM-inducible. Further characterization of *CYP81A-5A* will improve our understanding of what herbicide characteristics determine its potential of being a CYP81A substrate. This information could be utilized when determining herbicide rotations that minimize the chance for selecting grass weeds possessing CYP81As that are capable of detoxifying multiple different herbicides.

## Supporting information

**S1 Fig. Results of agriGO v2.0 related to transferase activity.** The color of the box indicates the significance level of the false discovery rate (reported in parentheses), with the yellow indicating relatively low significance and the gradation intensifies towards red to indicate higher significance. At the bottom of each significant box, the first fraction represents the number of significant differentially expressed genes assigned the specified GO term (out of 99), and the second fraction indicates the number of genes in the *Triticum aestivum* L. reference background with the same GO annotation.
(TIF)

**S2 Fig. Results of agriGO v2.0 related to oxidoreductase activity.** The color of the box indicates the significance level of the false discovery rate (reported in parentheses), with the yellow indicating relatively low significance and the gradation intensifies towards red to indicate higher significance. At the bottom of each significant box, the first fraction represents

the number of significant differentially expressed genes assigned the specified GO term (out of 99), and the second fraction indicates the number of genes in the *Triticum aestivum* L. reference background with the same GO annotation.
(TIF)

**S3 Fig.  Mean fold inductions of significant cytochrome P450s (CYPs) and UDP-dependent glucosyltransferase (UGTs) located on the group 5 wheat chromosomes.** Green bars represent UGTs and blue bars represent CYPs. Genes were induced by 15 g a.i. ha$^{-1}$ of cloquintocet-mexyl relative to untreated controls. Error bars indicate standard error of the mean.
(TIF)

**S4 Fig.  Mean cycle threshold (Ct) value for *β-tubulin* (*β-TUB*), *CYP81A-5A*, *CYP81A-5B*, and *CYP81A-5D* at 3, 6, and 12 hours after treatment (HAT) in untreated (UT) samples.** The UT treatment consisted of 0.1% nonionic surfactant, which is an adjuvant that was included in all other treatment utilized for this experiment. Within each timepoint, means that share the same letter are not significantly different (Fisher's LSD $\alpha = 0.05$). Mean Ct values represent results from three biological replicates ($n = 3$). Error bars represent standard error of the mean.
(TIF)

**S1 Table.  Reference and target gene primers and probes utilized for TaqMan RT-qPCR experiment.**
(XLSX)

**S2 Table.  Predicted amplified sequences of TaqMan primers and probes.** The underlined sequences correspond to the binding sites of the forward and reverse primers, respectively. Underlined and bolded sequences correspond to the binding sites of the TaqMan probes. Sequences are based on 'Chinese Spring' reference genome (IWGSC RefSeq v1.1) and functional gene annotations were downloaded from URGI (https://wheat-urgi.versailles.inra.fr/).
(XLSX)

**S3 Table.  All differentially expressed genes (DEGs) with Functional Annotations and Expression Data Sorted by Highest Fold Change (FC).**
(XLSX)

**S4 Table.  All GO Terms for each differentially expressed genes (DEGs).**
(XLSX)

**S5 Table.  List of significant differentially expressed genes (DEGs) identified by RNA-Seq and genes sharing the same functional annotation tandemly located on the same chromosome.**
(XLSX)

**S1 File.  Data associated with RT-qPCR experiments.**
(XLSX)

## Acknowledgments

We thank Dr. Alvaro Hernandez and Dr. Jenny Drnevich of the Roy J. Carver Biotechnology Center for providing technical advice for the construction of RNA-Seq libraries and statistical analysis. We also thank Dr. Kris N. Lambert, Dr. Norbert Satchivi and Dr. Carla Yerkes for providing technical support.

## Author contributions

**Conceptualization:** Olivia A. Landau, Dean E. Riechers.

**Data curation:** Olivia A. Landau.

**Formal analysis:** Olivia A. Landau, Brendan V. Jamison.

**Funding acquisition:** Dean E. Riechers.

**Investigation:** Olivia A. Landau, Brendan V. Jamison.

**Methodology:** Olivia A. Landau, Brendan V. Jamison.

**Project administration:** Dean E. Riechers.

**Resources:** Dean E. Riechers.

**Supervision:** Dean E. Riechers.

**Visualization:** Olivia A. Landau, Brendan V. Jamison.

**Writing – original draft:** Olivia A. Landau, Dean E. Riechers.

**Writing – review & editing:** Olivia A. Landau, Dean E. Riechers.

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
