## [Decision Letter · Decision Letter 0]

4 Dec 2024

PONE-D-24-43694Transcriptomic analysis reveals cloquintocet-mexyl-inducible genes in hexaploid wheat (Triticum aestivum L.)PLOS ONE

Dear Dr. Landau,

Thank you for submitting your manuscript to PLOS ONE. After careful consideration, we feel that it has merit but does not fully meet PLOS ONE’s publication criteria as it currently stands. Therefore, we invite you to submit a revised version of the manuscript that addresses the points raised during the review process.

We look forward to receiving your revised manuscript.

Kind regards,

Bahram Heidari

Academic Editor

PLOS ONE

“This research was supported by an Undergraduate Research Award at the University of Illinois Urbana-Champaign to B.V.J. and funding from Corteva Agriscience to D.E.R.”

Reviewers' comments:

Reviewer's Responses to Questions

**Comments to the Author**

1. Is the manuscript technically sound, and do the data support the conclusions?

Reviewer #1: Yes

Reviewer #2: Yes

2. Has the statistical analysis been performed appropriately and rigorously? 

Reviewer #1: Yes

Reviewer #2: Yes

3. Have the authors made all data underlying the findings in their manuscript fully available?

Reviewer #1: Yes

Reviewer #2: Yes

4. Is the manuscript presented in an intelligible fashion and written in standard English?

Reviewer #1: Yes

Reviewer #2: Yes

5. Review Comments to the Author

Reviewer #1: Dear authors

The authors have done interesting research entitled “Transcriptomic analysis reveals cloquintocet-mexyl-inducible genes in hexaploid wheat (Triticum aestivum L.)”. They have performed transcriptome sequencing of the wheat to find the inducible DEGs involved in some herbicide metabolisms. They have identified candidate genes responsible for halauxifen acid-detoxification and comprehensively discussed the related molecular mechanisms. They have shown that a novel member of the CYP81A subfamily of cytochrome P450s (CYPs) (CYP81A-5A) is capable of catalyzing synthetic auxin detoxification. This finding can be used in the genetic engineering of new herbicide-resistant wheat lines.

I believe that this manuscript is worthy to be accepted.

Regards,

Reviewer #2: The authors conducted a study to identify halauxifen acid (HA) detoxification genes using RNA-Seq. They compared untreated leaf tissue with leaf tissue treated with Cloquintocet-mexyl (CM). Through this analysis, they discovered the specific expression of CYP81A-5A and its homologues (CYP81A-5B and CYP81A-5D) in untreated leaf tissue, as well as in leaf tissue treated with CM and HM over time. While the findings of the study are innovative, further research is needed to fully support the claim that CYP81A-5A plays a role in herbicide detoxification. The results section should provide more detailed information on the number of differentially expressed genes (DEGs) and the gene ontology (GO) terms associated with transferase and oxidoreductase activity.

In summary, this study sheds light on potential HA detoxification genes, but additional studies are necessary to confirm the role of CYP81A-5A in this process.

Abstract:

Authors should provide a more detailed abstract regarding the number of reads that mapped to genes, as well as gene ontology analysis of differentially expressed genes.

Material and methods:

1. Please ensure that the names of all chemicals, the company name, and the percentage purity for each chemical are included in the Materials and Methods section of the manuscript.

2. The wording of line 98 should be formalize.

3. During the seedling stage, how was the plant irrigation performed before the treatment was administered?

4. When an abbreviation is used for the first time, it is not necessary to use the full name for subsequent mentions.

5. In line 120, why only one biological replicate per treatment per timepoint from each experimental replication was used for TaqMan RT-qPCR. It is advisable to include 3-5 biological replicates for each experimental condition in order to accurately measure experimental variation and establish the statistical significance of results. Additionally, it is recommended to conduct at least 3 technical replicates for each cDNA sample to assess assay variability and reduce the likelihood of pipetting errors.

Results:

1. Line 192, add the number of genes found through gene annotation that belong to the following categories: annotations, herbicide/xenobiotic metabolism, stress/defense response, amino acid metabolism, heavy metal binding, and encoded transcription factors. Specifically, indicate which genes were upregulated and which were downregulated in the category of herbicide/xenobiotic metabolism.

2. Line 195, The names of repressed genes need to be included.

3. In line 196, the author stated, "CM represses a few stress-related genes and likely triggers some level of oxidative stress, possibly to fine-tune safener-mediated transcriptional regulation." Author should be providing clarification on the number of genes that were suppressed.

4. The end of sentence of line 220 should be removed and relocated to the Materials and Methods.

5. In line 221, the author should specify the number of Gene Ontology (GO) terms associated with transferase activity, including both glutathione S-transferases (GST) and UDP-glucuronosyltransferases (UGTs), as well as oxidoreductase activity.

6. Line 236 should be shifted to M&M.

7. Which plant is involved with stress responses in UGT85As at line 247? Please include the name of the plant at the end of the sentence.

8. In line 257, the author stated that, "With the exception of TraesCS5A02G472300, these genes are located in tandem gene clusters encoding similar enzymes (Table S5)." Provide clarification on which specific genes are being referred to in this statement?

9. Line 271 should be relocated to the Methodology section of the report, as it is relevant to the methodology employed in the study, rather than the results obtained.

Discussion:

The discussion is too lengthy. It would be beneficial to eliminate redundant and overly informative sentences. For example, the sentence "Results of previous LC-MS [35] and phenotypic experiments [34] indicate that CYP81A-5D alone..." is repeated multiple times throughout the manuscript in the results, discussion, and conclusion sections. It is recommended to streamline the content and avoid unnecessary repetition for better clarity and conciseness.

Conclusion:

It is not customary to include references in the conclusion. Please remove them and instead, highlight the key findings of the study in the conclusion. Lines 415-420 should be shortened, and it is recommended that authors bold their key results for emphasis.

Figures:

- The red boxes in Supplementary Figures S1 and S2 were not sharpened.

- The supplementary figure S2 was not sharpened.

- In Supplementary Figure S4, there are errors in the significant letters for the 12- hours after treatment. Make the necessary corrections.

6. PLOS authors have the option to publish the peer review history of their article (what does this mean? ). If published, this will include your full peer review and any attached files.

**Do you want your identity to be public for this peer review?** For information about this choice, including consent withdrawal, please see our Privacy Policy .

Reviewer #1: **Yes: ** Ali Moghadam

Reviewer #2: **Yes: ** Maryam Salami

---

## [Author Response · Author response to Decision Letter 1]

13 Jan 2025

Revised Manuscript: PONE-D-24-43694

Responses to Comments from Reviewers

Reviewer #1: Dear authors

The authors have done interesting research entitled “Transcriptomic analysis reveals cloquintocet-mexyl-inducible genes in hexaploid wheat (Triticum aestivum L.)”. They have performed transcriptome sequencing of the wheat to find the inducible DEGs involved in some herbicide metabolisms. They have identified candidate genes responsible for halauxifen acid-detoxification and comprehensively discussed the related molecular mechanisms. They have shown that a novel member of the CYP81A subfamily of cytochrome P450s (CYPs) (CYP81A-5A) is capable of catalyzing synthetic auxin detoxification. This finding can be used in the genetic engineering of new herbicide-resistant wheat lines.

I believe that this manuscript is worthy to be accepted.

Regards,

Our response: We thank Reviewer #1 for their kind comments and for taking the time to review our manuscript.

Reviewer 2: Dear editor,

The authors conducted a study to identify halauxifen acid (HA) detoxification genes using RNA-Seq. They compared untreated leaf tissue with leaf tissue treated with Cloquintocet-mexyl (CM). Through this analysis, they discovered the specific expression of CYP81A-5A and its homologues (CYP81A-5B and CYP81A-5D) in untreated leaf tissue, as well as in leaf tissue treated with CM and HM over time. While the findings of the study are innovative, further research is needed to fully support the claim that CYP81A-5A plays a role in herbicide detoxification. The results section should provide more detailed information on the number of differentially expressed genes (DEGs) and the gene ontology (GO) terms associated with transferase and oxidoreductase activity.

In summary, this study sheds light on potential HA detoxification genes, but additional studies are necessary to confirm the role of CYP81A-5A in this process.

Our response: We thank Reviewer #2 for their kind comments and for taking the time to review our manuscript and providing constructive feedback. These comments were very helpful in improving the overall quality of the manuscript. The specific comments are addressed below.

Abstract:

Authors should provide a more detailed abstract regarding the number of reads that mapped to genes, as well as gene ontology analysis of differentially expressed genes.

Our response: More details regarding number of reads that mapped to genes and gene ontology analysis have been added to the abstract.

Material and methods:

1. Please ensure that the names of all chemicals, the company name, and the percentage purity for each chemical are included in the Materials and Methods section of the manuscript.

Our response: Additional chemical information has been added to the Materials and Methods.

2. The wording of line 98 should be formalize.

Our response: Edits to formalize the language have been made to the referenced line.

3. During the seedling stage, how was the plant irrigation performed before the treatment was administered?

Our response: Information about watering was added to the Materials and Methods.

4. When an abbreviation is used for the first time, it is not necessary to use the full name for subsequent mentions.

Our response: We noticed this issue with the term “gene ontology” and made corrections in the Materials and Methods and Results sections. Please let us know if there are other issues that were missed. We did notice previously defined abbreviations being defined again for the figure captions. If this is the source of the issue, we feel that redefining the abbreviations in the figure captions is helpful to readers and allows these figures to better stand on their own without having to read the manuscript.

5. In line 120, why only one biological replicate per treatment per timepoint from each experimental replication was used for TaqMan RT-qPCR. It is advisable to include 3-5 biological replicates for each experimental condition in order to accurately measure experimental variation and establish the statistical significance of results. Additionally, it is recommended to conduct at least 3 technical replicates for each cDNA sample to assess assay variability and reduce the likelihood of pipetting errors.

Our response: In total, we used 3 independent biological replicates per time point per treatment. This means each biological replicate came from an independent experiment, which is same procedure that was utilized for our RNA-Seq experiment. We believe this is the best way to capture variation and demonstrate repeatability of the results. Given that PLOS ONE does not have guidelines on the number of biological replicates or whether these replicates are from the same or different independent experimental replications, we believe that our methods meet the standards of PLOS ONE. Please see several recent research articles from PLOS ONE, cited below, that highlight variation in methods for biological replicate utilization. Further emphasis on the total number of biological replicates that were utilized in RNA-Seq and TaqMan RT-qPCR were added to Materials and Methods. Three technical replicates were utilized for each sample, which is noted in line 194. This is an appropriate area to mention this aspect because it is among the other details pertaining to how RT-qPCR was performed.

Zhu S, Chen L, Zhang Z, Chen G, Hu N (2024) BnVP1, a novel vacuolar H+ pyrophosphatase gene from Boehmeria nivea confers cadmium tolerance in transgenic Arabidopsis. PLOS ONE 19(8): e0308541. https://doi.org/10.1371/journal.pone.0308541

Cheaib M, Nguyen HT, Couderc M, Serret J, Soriano A, et al. (2024) Transcriptomic and metabolomic reveal OsCOI2 as the jasmonate-receptor master switch in rice root. PLOS ONE 19(10): e0311136. https://doi.org/10.1371/journal.pone.0311136

Bhattarai G, Rhein HS, Sreedasyam A, Lovell JT, Khanal S, et al. (2024) Transcriptome analysis under pecan scab infection reveals the molecular mechanisms of the defense response in pecans. PLOS ONE 19(11): e0313878. https://doi.org/10.1371/journal.pone.0313878

Results:

1. Line 192, add the number of genes found through gene annotation that belong to the following categories: annotations, herbicide/xenobiotic metabolism, stress/defense response, amino acid metabolism, heavy metal binding, and encoded transcription factors. Specifically, indicate which genes were upregulated and which were downregulated in the category of herbicide/xenobiotic metabolism.

Our response: Additional details about the number of genes upregulated and downregulated have been added to Results.

2. Line 195, The names of repressed genes need to be included.

Our response: Names of the genes were added to Results.

3. In line 196, the author stated, "CM represses a few stress-related genes and likely triggers some level of oxidative stress, possibly to fine-tune safener-mediated transcriptional regulation." Author should be providing clarification on the number of genes that were suppressed.

Our response: The number of repressed genes were added to Results.

4. The end of sentence of line 220 should be removed and relocated to the Materials and Methods.

Our response: This line was removed and information was relocated to lines the Materials and Methods.

5. In line 221, the author should specify the number of Gene Ontology (GO) terms associated with transferase activity, including both glutathione S-transferases (GST) and UDP-glucuronosyltransferases (UGTs), as well as oxidoreductase activity.

Our response: After reading this section, we identified an error in our wording: we meant to describe the significant number of genes associated with the GO terms and not the significant number of GO terms associated with the genes. Information regarding the significant number of genes associated with the specified GO terms has been added to Results.

6. Line 236 should be shifted to M&M.

Our response: This line was removed and information was relocated to lines to Materials and Methods. The citation numbers were also adjusted accordingly.

7. Which plant is involved with stress responses in UGT85As at line 247? Please include the name of the plant at the end of the sentence.

Our response: Requested information was added to the Results.

8. In line 257, the author stated that, "With the exception of TraesCS5A02G472300, these genes are located in tandem gene clusters encoding similar enzymes (Table S5)." Provide clarification on which specific genes are being referred to in this statement?

Our response: Requested information about specific genes located in clusters was added to the Results.

9. Line 271 should be relocated to the Methodology section of the report, as it is relevant to the methodology employed in the study, rather than the results obtained.

Our response: Information was relocated to lines to the Materials and Methods.

Discussion:

The discussion is too lengthy. It would be beneficial to eliminate redundant and overly informative sentences. For example, the sentence "Results of previous LC-MS [35] and phenotypic experiments [34] indicate that CYP81A-5D alone..." is repeated multiple times throughout the manuscript in the results, discussion, and conclusion sections. It is recommended to streamline the content and avoid unnecessary repetition for better clarity and conciseness.

Our response: Redundant phrases were removed and wording was altered throughout the discussion section to be more concise.

Conclusion:

It is not customary to include references in the conclusion. Please remove them and instead, highlight the key findings of the study in the conclusion. Lines 415-420 should be shortened, and it is recommended that authors bold their key results for emphasis.

Our response: References were removed and the entire conclusion was shortened. We did not bold key findings because we do not believe this is appropriate and it is not a requirement by PLOS ONE. Some information was relocated to the Discussion because we believe this information was worth noting and better suited for this section.

Figures:

- The red boxes in Supplementary Figures S1 and S2 were not sharpened.

- The supplementary figure S2 was not sharpened.

- In Supplementary Figure S4, there are errors in the significant letters for the 12- hours after treatment. Make the necessary corrections.

Our response: Supplementary Figures S1 and S2 have been sharpened. We don’t see any errors in Supplementary Figure S4. Further elaboration on the errors is needed if something specific needs to be changed.

---

## [Decision Letter · Decision Letter 1]

29 Jan 2025

Transcriptomic analysis reveals cloquintocet-mexyl-inducible genes in hexaploid wheat (Triticum aestivum L.)

PONE-D-24-43694R1

Dear Dr. Landau,

We’re pleased to inform you that your manuscript has been judged scientifically suitable for publication and will be formally accepted for publication once it meets all outstanding technical requirements.

Kind regards,

Bahram Heidari

Academic Editor

PLOS ONE

Additional Editor Comments (optional):

Reviewers' comments:

Reviewer's Responses to Questions

**Comments to the Author**

1. If the authors have adequately addressed your comments raised in a previous round of review and you feel that this manuscript is now acceptable for publication, you may indicate that here to bypass the “Comments to the Author” section, enter your conflict of interest statement in the “Confidential to Editor” section, and submit your "Accept" recommendation.

Reviewer #2: (No Response)

2. Is the manuscript technically sound, and do the data support the conclusions?

Reviewer #2: Yes

3. Has the statistical analysis been performed appropriately and rigorously? 

Reviewer #2: Yes

4. Have the authors made all data underlying the findings in their manuscript fully available?

Reviewer #2: Yes

5. Is the manuscript presented in an intelligible fashion and written in standard English?

Reviewer #2: Yes

6. Review Comments to the Author

Reviewer #2: (No Response)

7. PLOS authors have the option to publish the peer review history of their article (what does this mean? ). If published, this will include your full peer review and any attached files.

**Do you want your identity to be public for this peer review?** For information about this choice, including consent withdrawal, please see our Privacy Policy .

Reviewer #2: **Yes: ** Maryam Salami

---

## [Editor Report · Acceptance letter]

PONE-D-24-43694R1

PLOS ONE

Dear Dr. Landau,

I'm pleased to inform you that your manuscript has been deemed suitable for publication in PLOS ONE. Congratulations! Your manuscript is now being handed over to our production team.

Kind regards,

on behalf of

Dr. Bahram Heidari

Academic Editor

PLOS ONE